# Clinical Research in Chronic Lymphocytic Leukemia in Pakistan; A Systematic Review

**DOI:** 10.3390/medicina59081483

**Published:** 2023-08-17

**Authors:** Mohammad Ammad Ud Din, Moazzam Shahzad, Aqsa Ashraf, Hania Liaqat, Ali Jaan, Faiz Anwer

**Affiliations:** 1Hematology/ Oncology, Moffitt Cancer Center, University of South Florida, 12902 USF Magnolia Drive, Tampa, FL 33612, USA; 2Hematology/ Oncology, University of South Florida, Tampa, FL 33612, USA; 3Internal Medicine, Northwell Health Mather Hospital, Port Jefferson, New York, NY 11777, USA; 4Internal Medicine, Rochester General Hospital, Rochester, New York, NY 14621, USA; 5Hematology/ Oncology, Cleveland Clinic, Cleveland, OH 44195, USA

**Keywords:** blood malignancies, chronic lymphocytic leukemia, malignant hematology, Pakistan, small cell leukemia

## Abstract

*Background*: Significant advances have been made in the treatment of chronic lymphocytic leukemia (CLL) since the turn of the new millennium. However, most clinical trials were done in developed countries where minority ethnicities were underrepresented. *Materials and Methods*: To gauge the quality of research in CLL being done in Pakistan, we conducted a comprehensive literature search using PubMed, Clinicaltrials.gov, and Google Scholar on 14 January 2022 following the Preferred Reporting Items for Systematic Reviews and Meta-Analyses (PRISMA) recommendations. *Results*: A total of 16 studies met the inclusion criteria. The most common study design was cross-sectional. Eight studies evaluated the clinicohematological profile of CLL patients and the effect of various cytogenic abnormalities through fluorescence in situ hybridization (FISH) technique on disease progression and prognosis. Five studies discussed the prevalence of abnormalities such as autoimmune cytopenias and other serum chemistry derangements. Only two studies evaluated treatment outcomes, among which one study reported a 2-year overall survival of 65% among patients with 17p deletion. None of the studies had patients on novel targeted agents. No pharmaceutical sponsored or funded clinical trials were found. *Conclusions*: Our review suggests that although small clinical studies continue to be performed across the country, multiple financial and logistical barriers need to be addressed for larger, more impactful clinical trials to be conducted that will help answer demographic-specific questions and decrease reliance on foreign studies.

## 1. Introduction

Chronic lymphocytic leukemia (CLL) is the most common adult leukemia worldwide. It is largely a disease of the elderly with the mean age of diagnosis around 72 years [1]. The B-cell malignancy is characterized by the accrual of mature monoclonal CD-5+ B cells in the bone marrow, lymphoid organs, and/or peripheral bloodstream [2]. More than a quarter of the patients are asymptomatic at the time of diagnosis and the clinical course can be highly heterogeneous ranging from indolent disease never requiring treatment to aggressive disease refractory to multiple lines of therapy [1,2]. The incidence of the malignancy shows remarkable global variation accounting for more than 30% of all leukemias in Western countries, while less than 5% of leukemias in the Far East, likely due to underlying genetic predisposition in the white population as the incidence of CLL is low in Asian immigrants living in North America and Europe [3]. In Pakistan, the incidence of the disease is relatively low, but in the absence of a national cancer registry, the true incidence is likely underreported. Nonetheless, the incidence of CLL is increasing in developing countries and the data regarding overall survival (OS) and treatment strategies are not abundantly reported and are heavily dependent on the socioeconomic settings and access to quality healthcare [3]. Over the last two decades, the treatment for CLL has undergone a paradigm shift from conventional chemotherapeutic agents such as chlorambucil to more targeted therapies such as Bruton tyrosine kinase inhibitors (BTKi) and BCL-2 inhibitors such as ibrutinib and venetoclax, respectively, with or without anti-CD20 antibodies such as rituximab [4,5]. These agents have proven to be significantly beneficial in terms of inducing deep remissions and improving quality of life due to a more tolerable adverse-effect profile. In addition, the multiple genetic mutations through the use of next generation sequencing (NGS) have been identified that have improved our ability to identify patients with high-risk diseases and predict their response to treatment, allowing clinicians to risk-stratify patients and move away from a ‘one size fits all’ model of treatment [4,5]. As most of the studies regarding the advancements in CLL have been conducted in the developed world, minority ethnicities were under-represented in large clinical trials. Therefore, quality local research is important to answer demographic-specific questions, reduce the reliance on foreign research, and bridge socio-cultural differences to address the needs of native patients. With this premise, we conducted a systematic review of the literature to examine and summarize the best published clinical research regarding CLL from Pakistan. The decision to choose clinical studies from Pakistan was based on multifold reasons, including (i) all authors were previously involved in both clinical and academic activities in Pakistan allowing them to recognize and comment on the quality of research conducted in the country. (ii) In recent years, there has been much interest in addressing the healthcare disparities in cancer care and the authors believe a review of international studies that included Pakistan as a study site would help summarize the international efforts done so far and identify areas where additional work is needed. (iii) The discipline of malignant hematology was only formally recognized as a separate specialty in medicine in the early 1990s in Pakistan. As scholarly activities in the field started less than 30 years ago, this provided authors with a tangible timeframe and the reassurance that all major local studies published to date would be included in the review from a remote Internet-based literature search strategy as all journal archives accessed had their prior volumes available online.

## 2. Materials and Methods

Three large databases, namely PubMed, Clinicaltrials.gov, and Google Scholar, were accessed from Rochester, New York to conduct a comprehensive electronic literature search from the inception of the databases till 14 January 2022. Studies published in both local and international journals were included. The search strategy is provided in Appendix A. The articles retrieved were initially imported into EndNote 20 and screened for removal of any duplicate results. The remaining articles were then further screened by two independent reviewers (M.A., and H.L.), first by titles and abstracts, followed by a full-text review of filtered articles to determine eligibility for inclusion. Any disputes were resolved through mutual discussion. Only clinical studies performed at a center in Pakistan specifically evaluating CLL patients were included. All other articles including case reports, editorials, national survey reports, basic science studies, and studies related to any other malignancy were excluded. This systematic review was performed following the Preferred Reporting Items for Systematic Reviews and Meta-Analyses (PRISMA) recommendations. The PRISMA flow chart for the inclusion of studies is presented in Figure 1 [6].

## 3. Results

A total of 16 studies met the inclusion criteria (n = 1264 patients) [7,8,9,10,11,12,13,14,15,16,17,18,19,20,21,22]. The summary of the included studies is provided in Table 1 and Table 2. The most common study design was cross-sectional (n = 13), two studies were prospective, and one was retrospective in nature. The mean sample size was 79 ± 35 patients. The mean age across 13 of the included studies was 59 ± 6 years. Eight studies described the general hematologic and clinical profile of CLL patients who sought care at their respective institutes [7,8,9,10,11,12,13,14]. The authors reported the prevalence of abnormalities noted in the patients’ complete blood profiles such as anemia, thrombocytopenia, and elevated total leukocyte count. Rafiq et al. also demonstrated an increased incidence of elevated liver enzymes and serum creatinine in CLL patients, though a correlation between these anomalies and CLL could not be established as the pre-existing co-morbidities of the patients were not accounted for [8]. Abbas et al. [9] assessed the incidence of autoimmune cytopenias and reported a Coomb’s positivity rate of 23.3% while Rashid et al. reported a slightly higher rate of 26.7% [12]. Understandably, the Coomb’s positive patients had a higher-grade disease and were more thrombocytopenic than their Coomb’s negative counterparts. About 20% of CLL patients studied by Ehsan et al. [7] developed autoimmune hemolytic anemia while Haider et al. reported a much lower incidence of 7.8% [13]. Both author groups reported a similar 3% rate of immune thrombocytopenic purpura (ITP) in their study population. The severity of the cytopenias was not discussed in either study and the percentage of patients requiring hospitalization for treatment or transfusion needs was not disclosed. A study by Parveen et al. concluded that more than half of CLL patients suffer from Vitamin D deficiency; however, no statistically significant correlation was found between Vitamin D deficiency and disease severity or prognostic biomarkers [10]. The use of fluorescent in situ hybridization (FISH) to detect cytogenic abnormalities for risk-stratification in CLL has become more widespread in Pakistan and recently multiple studies have assessed the prognostic significance of various such markers in the local population. TP53 mutation and 17p deletion are indicators of rapid disease progression and poor treatment response. Qadir et al. evaluated 139 CLL patients to determine the frequency of TP53 gene mutation and reported a prevalence of 13.7% among their study pool [17]. Age and gender were not found to have a statistically significant correlation with TP53 mutation. Mahmood et al. compared 24 patients with 17p deletion to 106 controls and demonstrated a statistically significant increase in B symptoms (*p* = 0.007) and elevated beta-2 microglobulin levels (*p* = 0.001) among the 17p deletion groups [20]. These patients were also significantly older with comparatively higher disease than the control population (*p* = 0.001 and *p* = 0.01 respectively). The 2-year OS was 65% among deletion 17p and was not statistically significant compared to patients without 17p deletion. Another cytogenetic abnormality commonly associated with CLL is the deletion of 11q22. This causes resistance to DNA destructive therapy resulting in a poor prognosis. Korejo et al. studied 61 patients with CLL and found 7 patients (11.4%) with 11q22 deletion [18]. The comparison of two groups with and without 11q22 deletion indicated a statistically insignificant difference between age, organomegaly, or Rai staging. Ahmed et al. studied 56 patients with CLL to determine the impact of trisomy 12 on prognosis and concluded that the chromosomal anomaly did not affect staging, 1-year disease progression, or chemotherapy dependence [16]. Trisomy 12 was otherwise detected in 12 patients (10.7%). Ahmed et al. also assessed the effect of 13q14 deletion in 21 out of 56 CLL patients (37.5%) [15]. Only 23.8% patients with 13q14 had disease progression at 1-year follow-up compared to 68.7% of the controls RR: 0.34 (0.15–0.77); *p* = 0.001. Zeeshan et al. reported that 13.5% of 89 CLL patients in their expressed ZAP-70 positivity and these patients had a higher absolute lymphocyte count and advanced clinical disease at the time of disease presentation compared to ZAP-70 negative patients [22]. In contrast, among 101 patients studied by Khaliq et al., only 3% expressed ZAP-70. They also reported that 33% of patients expressed CD38 among whom 92% had stage III and IV disease at the time of diagnosis [19]. Nazir et al. compared various first-line treatments for CLL including FCR (fludarabine, cyclophosphamide, rituximab) in 5 patients (11%), FC in 38 patients (83%), chlorambucil in 2 patients (4%), and CVP (cyclophosphamide, vincristine, prednisone) in 1 patient (2%). The study group had a generally favorable outcome as 22 patients (56%) achieved complete response, 13 (33%) had a partial response, 3 (7.6%) had stable disease, and 1 (2.5%) patient had disease progression. More relapses were seen in the FC group as compared to the FCR group and the authors concluded that the addition of rituximab to the regimen results in improved outcomes [21].

## 4. Discussion

Although the incidence of CLL in Pakistan is much lower compared to other malignant blood disorders according to regional reports, the true incidence rate is unknown due to the absence of a national registry [23]. Overall, in the South Asian region, the incidence of CLL has more than tripled from 1990 to 2019 with the age-standardized incidence rate (ASIR) reaching 0.54 per 100,000 persons in 2019 [24]. Since most of the patients are asymptomatic at diagnosis and access to regular blood count screening and immunophenotyping is not widely available, CLL is likely underdiagnosed in Pakistan, particularly in the rural districts. Similar diagnostic barriers were also reported in studies from northern India and Sudan [25,26]. From the studies included in our review, the mean age of diagnosis of CLL is about a decade earlier in the Pakistani population compared to the West. Though evidence in the form of quality cohort studies is lacking, global epidemiologic studies have hypothesized that the increased exposure to certain carcinogens such as benzene and formaldehyde in developing countries may be a contributory factor to the early onset of disease [24]. Furthermore, compared to the West, the use of tobacco products remains high in countries like Pakistan. Tobacco use has been shown to increase the risk for hematologic malignancies [24]. While many CLL patients often do not need therapy at the time of diagnosis, the early onset leads to ‘added years’ for various immunologic complications to manifest. Our review demonstrates that studies have been performed to determine the prevalence of autoimmune cytopenias, but no study has been done so far to evaluate the immune status of CLL patients in Pakistan. This is an important area of research as hypogammaglobulinemia is common, particularly in patients with high-grade disease, and predisposes them to recurrent infections [27,28]. There is no specific literature regarding the pattern of infections in Pakistani CLL patients, but in general, vaccinations remain the cornerstone in preventing infections. Every effort should be made to vaccinate these patients against respiratory infections such as streptococcal pneumonia [27,28]. This is of paramount importance, especially during the pandemic as studies have shown CLL patients are at a higher risk of developing severe COVID-19 [29,30]. In addition to vaccines, the literature also supports the initiation of immunoglobulin replacement therapy (IgR) in individuals with low IgG levels and/or recurrent infections for infection prophylaxis to decrease the risk of severe infections [27]. IgR is often administered intravenously and requires specialized infusion centers that often lead to a significant increase in the cost of treatment. Moreover, in resource-constrained countries such as Pakistan, there are not enough infusion centers to meet the needs of the patients which further makes the provision of comprehensive care for CLL patients difficult. IgR can also be self-administered subcutaneously at home by patients [27], but this is not a common practice and currently unavailable in Pakistan.

Although multiple studies have been performed to evaluate the prevalence of various biochemical prognostic markers, none of these studies discussed whether the presence of certain high-risk cytogenetics such as TP53 mutation or 17p deletion influenced the treatment regimen offered to the patients if they met clinical criteria for initiation of treatment. Qadir and colleagues reported the prevalence of TP53 mutation at around 14% in their study population, which is nearly double compared to studies done in the West [17,31]. Studies from India have also reported a prevalence of TP53 mutation in up to 10% of CLL patients at diagnosis [32]. Similarly, the prevalence of 17p deletion was 18% in the study by Mahmood and colleagues whereas a single-center study from India reported the prevalence of 11.4% in treatment-naïve patients [20,33]. Although these studies did not have a large enough sample size to allow population-level generalizability of the findings, they do pose the question of whether the prevalence of CLL with high-risk cytogenetics is higher in South Asia in comparison to the developed world. Additionally, because most of these studies were cross sectional studies, the OS of patients based on differences in treatment or cytogenetics was not assessed and there is a paucity of data in terms of OS of CLL patients. Only Mahmood et al. reported a 2-year OS of 65% among 24 patients with 17p deletion. Interestingly, there was no significant difference in OS between patients with and without 17p deletion on univariate analysis accounting for age, beta 2 microglobulin level, ZAP 70 status, Binet staging, B symptoms, or 17p deletion clone size. This was likely because of the small study population with 16% of patients among the 24 patients with 17p deletion lost from follow-up early in the course. As the study participants were enrolled between 2013 and 2015, treatment regimens included chlorambucil, FCR, bendamustine and rituximab, and glucocorticoids alone where 17p did not impact the treatment choice. Novel agents such as ibrutinib have shown to be effective in managing patients with high-risk cytogenetics, but there are limited data available regarding their use in South Asian countries. Clinical studies from Pakistan related to therapeutics in CLL are scarce. Apart from the earlier discussed study [20], our review identified only one additional observational study done with the goal of evaluating the treatment outcomes of CLL patients in Pakistan [20,21]. Neither of these studies had any patients on the novel targeted agents. Although some novel agents such as BTKi are available for use in Pakistan, no local institution has published their experience with the native population. In contrast, data regarding the use of BTKi and its impact on the quality of life for CLL patients are emerging from India [32,33,34]. Apart from clinical studies, there are no local financial reports published gauging the cost of treatment with these agents from Pakistan either. As there are no large national insurance programs in Pakistan, it is important to consider the economic burden of treatment when deciding between initiation of lifelong therapy with BTKi versus fixed-duration therapy with FCR as most patients pay out of pocket. Moreover, in the absence of universal health care, there is a lot of heterogeneity in terms of treatment options for CLL when comparing private and public health setups due to the cost of novel agents. One small Indian study adopted a nonconventional reduced dose of ibrutinib to lower the overall cost of treatment and demonstrated similar efficacy in terms of progression-free survival when compared with patients on standard dose [32]. Similarly, agents such as venetoclax are not available in government-funded hospitals but can be imported on request for patients being treated in private centers [35]. Even if novel agents can somehow be available at subsidized rates to the public in the future, BTKi therapies have significant adverse effects such as atrial fibrillation which add to the cost of care [36]. The detection of such arrhythmias might require additional interventions such as a Holter monitor or a Zio patch which are not only expensive, but their availability is also limited to a few cardiac centers. Given these unique population-specific dilemmas, studies evaluating the use of BTKi in Pakistan are needed to better understand the financial and logistic barriers associated with their widespread use.

Although clinical trials for solid tumors such as pancreatic and liver cancer [37,38] have been done, there have been no pharmaceutical-sponsored clinical studies for CLL done in Pakistan to date. All studies in our review were not funded and were mostly conducted by postgraduate trainees under the supervision of their respective mentors. The lack of funding for clinical studies is a longstanding barrier to quality research in the country [39]. As of 2017, Pakistan contributed to only 0.1% of all clinical trials done worldwide despite having a large patient pool with considerable avenues for research and state-of-the-art Joint Commission accredited centers. National-level trial registries, regulatory frameworks, and ethical committees are all essential components needed to foster an academic culture, and until a solid research infrastructure is established, the capacity to obtain funding to conduct randomized-controlled trials or long-term prospective studies remains limited [40]. While efforts are being made by national societies to accelerate the growth of such activities, global collaboration is critical in such endeavors. This issue stems beyond the scope of the medical fraternity alone and requires interest from the health, foreign, industrial, and production ministries of Pakistan to incentive pharmaceutical interest in Pakistan. Clinical trials can sometimes be the only opportunity for treatment using novel agents for patients from low-resource countries but unfortunately, the current situation is unlikely to change in the foreseeable future. Recently, international scientific conferences such as HAEMCON have helped improve international ties and address some of the roadblocks.

Even though the authors followed a systematic approach, gathered data from multiple databases, thoroughly reviewed the contents of all articles, and resolved conflicts through mutual discussion among all authors to determine the final decision on the eligibility of inclusion of the article into the review, there are certain limitations that need to be stated. Firstly, several Pakistani studies are published in local print only or locally indexed journals that may have not been captured by our online literature search. Secondly, regional health reports such as those published by the World Health Organization to determine worldwide epidemiological characteristics of lymphomas were excluded from our review. Thirdly and most importantly, due to significant heterogeneity among study participants, outcomes of interest, and patient data reported, it was difficult to pool data to draw clinically relevant inferences. Regardless of these limitations, we believe our review effectively highlights the landscape of clinical research on CLL in Pakistan.

## 5. Conclusions

As with all other malignancies, there is enormous potential for large-scale research in CLL in Pakistan. With multiple small single-center studies done so far, the future looks optimistic especially as more institutions dedicated to hematologic malignancies are established across the country. However, much work remains to be done to improve the local research culture and attract pharmaceutical companies to acquire international funding for industry-sponsored randomized-controlled trials that garner the interest of the international community. Future studies should incorporate the use novel diagnostic modalities such as NGS in risk stratification and evaluate treatment outcomes with targeted therapies along with their impact on cost of treatment and quality of life.

## Figures and Tables

**Figure 1 medicina-59-01483-f001:**
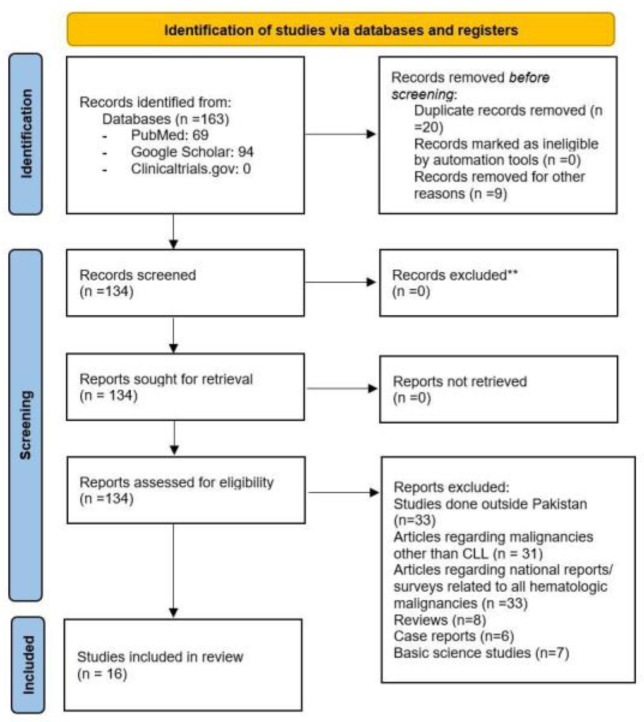
Screening criteria for the included studies according to the The PRISMA 2020 statement. **: Based on independent review of two authors.

**Table 1 medicina-59-01483-t001:** Summary of studies evaluating clinical and laboratory profile of CLL patients.

Author(s) Year	Sample Size (n)	Mean Age (Years)	Study Design	Study Objective	Results
Ehsan et al., 2013 [7]	31	62.8	Cross-sectional	Determine the prevalence of autoimmune cytopenias in CLL patients	Autoimmune cytopenias were seen in 22% of the patients. Autoimmune hemolytic anemia occurred more frequently compared to immune thrombocytopenia purpura (19.4% vs. 3.2%).
Rafiq et al., 2014 [8]	50	41.5 ± 20.86	Cross-sectional	Assess the clinico-hematological profile of CLL patients	CLL patients were more anemic and thrombocytopenic. creatinine, total bilirubin, and urea were also raised above normal limits.
Abbas et al., 2015 [9]	60	59 ± 9.2	Cross -sectional	Determine the correlation of Coombs test with disease staging	There was a statistically significant association between Coomb’s positivity and advanced Rai stage III disease and low mean hemoglobin.
Parveen et al., 2015 [10]	60	59.0 ± 9.2	Cross-sectional	Determine the prevalence of Vitamin D deficiency in CLL patients	Vitamin D insufficiency was found in 56.7% of the patients. No correlation was found with age, clinical stage, or other biochemical markers.
Zeeshan et al., 2015 [11]	60	59.0 ± 9.2	Prospective observational	Assess the clinico-hematological profile of CLL patients	Anemia and thrombocytopenia were seen in 26.7% and 21.7% of cases, respectively, and 21.7% had Rai Stage IV disease.
Rashid et al., 2018 [12]	150	65.8 ± 1.5	Cross-sectional	Gauge the frequency of hematological complications in CLL patients	Of the patients, 71% had at least some degree of thrombocytopenia.
Haider et al., 2019 [13]	64	65	Cross-sectional	Determine the prevalence of autoimmune cytopenias in CLL patients	Autoimmune hemolytic anemia occurred more frequently compared to immune thrombocytopenia purpura (7.8% vs. 3.1%).
Rashid et al., 2020 [14]	100	65.8 ± 1.33	Cross-sectional	Assess the frequency of autoimmune haemolytic anemia in CLL patients	Coomb’s positivity rate was 26.7%. Coomb’s positive patients had a mean hemoglobin of 7.69 g/dL ± 2.3.

**Table 2 medicina-59-01483-t002:** Summary of studies related to molecular prognostic markers or treatment outcomes.

Author(s) Year	Sample Size (n)	Mean Age (Years)	Study Design	Study Objective	Results
Ahmed et al., 2021 [15]	56	60	Cross-sectional	Determine the prevalence of 13q14 deletion and its effect on prognosis in CLL patients	13q14 deletion was present in 37.5% of the patients. Most of these patients had low-grade disease and low rate of disease progression.
Ahmed et al., 2021 [16]	56	60	Cross-sectional	Assess the clinicohematological characteristics of CLL patients with trisomy 12	Trisomy 12 was detected in 10.7% of the patients. There was no statistically significant association of trisomy 12 with aggressive disease or poor prognostic markers.
Qadir et al., 2020 [17]	139	56.3 ± 10.84	Cross-sectional	Determine the frequency of 17p deletion with TP53 mutation in CLL patients	TP53 mutation was found in 13.7% of the patients. Age and gender were not statistically significant with TP53 mutation.
Korejo et al., 2020 [18]	61	Not available	Cross-sectional	Assess the clinicohematological characteristics of CLL patients with 11q22 deletion	Deletion 11q22 was detected in 11.4% of the patients. No statistical significant correlation was found between 11q22 deletion and age or Rai staging.
Khaliq et al., 2019 [19]	101	64 *	Cross-sectional	Determine the frequency of Zap-70 and CD38 in newly diagnosed B cell CLL patients	2.97% of patients were positive for Zap-70 and 33.66% of newly diagnosed patients were positive for CD38.
Mahmood et al., 2018 [20]	130	64.2	Prospective observational	Assess the clinicohematological characteristics of CLL patients with 17p deletion	17p deletion was present in 18% of the patients and was associated with age > 65 years (*p* = 0.01), advanced clinical stage (*p* = 0.0001), presence of B symptoms (*p* = 0.007), and elevated B2 microglobulin levels (*p* = 0.001). No 2 year survival difference in patients with or without 17p deletion.
Nazir et al., 2016 [21]	57	50.9 *	Retrospective cohort study	To assess the outcomes with different chemotherapeutic regimens in patients with CLL	Regimens used were FC in 83%, FCR in 11%, chlorambucil in 4% and CVP in 2% of the patients. ORR was 89%. Median PFS was 23.1 months and median 3 years OS was 55%.
Zeeshan et al., 2015 [22]	89	57.5 ± 12.1	Cross-sectional	Determine the prevalence of ZAP-70 positivity in B-CLL patients at disease presentation	ZAP-70 positivity rate was 13.7% and had a statistically significant correlation with Rai Stage III disease and increased total lymphocyte count. No correlation established with age or gender.

*: Median age.

## Data Availability

No new data were created or analyzed in this study. Data sharing is not applicable to this article.

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
