# Peer review of "Clinical Research in Chronic Lymphocytic Leukemia in Pakistan; A Systematic Review"

_medicina, 2023, doi:10.3390/medicina59081483_

Round 1

Reviewer 1 Report

This is a very interesting report on CLL in Pakistan. 

I would like if the authors specify the average survival of general population  in Pakistan, becouse this could explain the low median age at diagnosis of CLL 

Author Response

Manuscript Title: Clinical Research in Chronic Lymphocytic Leukemia in Pakistan; A Systematic Review

Manuscript ID: medicina-2534481

Date: August 12th, 2023

Response to Reviewer# 2 Comments:

The authors sincerely thank the reviewer for taking the time to critically evaluate the integrity and merit of our manuscript. We have thoroughly revised our manuscript based on the feedback received. All of the reviewer’s comments have been addressed to the best of our capability. Please find below a point-by-point response to the reviewer comments:

Comment 1: I would like if the authors to specify the average survival of the general population  in Pakistan, because this could explain the low median age at diagnosis of CLL

Response: Though the authors agree with the reviewer that the overall survival of CLL patients in Pakistan is an important statistic to incorporate into the review, unfortunately, only 1 study reported the 2-year overall survival. This has been added to the results and discussion section.

Thank you again for considering and reviewing our manuscript. Please do not hesitate to reach out to me in case further revisions or clarifications are needed.

Regards,

Mohammad Ammad Ud Din, MD

Reviewer 2 Report

The manuscirpit discusses CLL clinical research in Pakistan, where it is relatively low in incidence. The review highlights the need for local research to address demographic-specific questions and bridge cultural differences. It includes 16 studies involving 1,264 patients, covering various aspects of CLL, such as hematologic profiles, cytogenetic abnormalities, and treatment outcomes.

-In the Introduction, elaborate on the significance of CLL as a disease and its impact on global health. Include more details about its prevalence and incidence worldwide to set the stage for why it is underdiagnosed and why therefore  CLL research in Pakistan is important.
-In the Results section, provide more detailed findings from the included studies. Discuss the statistical significance, clinical relevance, and implications of the results. Include specific numbers, percentages, and outcomes of interest for each study.

-In the Discussion section, interpret the results in the context of existing literature and previous research. Discuss in more details discrepancies or similarities with studies conducted in other regions. Highlight the contributions of the local research in Pakistan and its potential impact on patient care and management.

-Acknowledge the limitations of the review, such as the small sample sizes of some studies, potential biases, and the absence of certain types of research..

-Conclude the Discussion by suggesting future research areas that could expand on the existing findings and fill the gaps in knowledge about CLL in Pakistan. This could include investigating specific genetic factors, treatment outcomes, and potential strategies for improving patient care in the country.

Author Response

Manuscript Title: Clinical Research in Chronic Lymphocytic Leukemia in Pakistan; A Systematic Review

Manuscript ID: medicina-2534481

Date: August 12th, 2023

Response to Reviewer# 3 Comments:

The authors sincerely thank the reviewer for taking the time to critically evaluate the integrity and merit of our manuscript. We have thoroughly revised our manuscript based on the feedback received. All of the reviewer’s comments have been addressed to the best of our capability. Please find below a point-by-point response to the reviewer comments:

Comment 1: In the Introduction, elaborate on the significance of CLL as a disease and its impact on global health. Include more details about its prevalence and incidence worldwide to set the stage for why it is underdiagnosed and why therefore CLL research in Pakistan is important.

Response: The introduction has been elaborated. We have identified lack of access to healthcare and socioeconomic barriers as possible reasons for under diagnosis.

Comment 2: In the Results section, provide more detailed findings from the included studies. Discuss the statistical significance, clinical relevance, and implications of the results. Include specific numbers, percentages, and outcomes of interest for each study.

Response: Unfortunately, the bulk of the studies were descriptive with little to no statistical calculations performed. The authors have highlighted all key clinically relevant findings to the best of our abilities. Table 1 and 2 also summarizes these points.

Comment 3: In the Discussion section, interpret the results in the context of existing literature and previous research. Discuss in more details discrepancies or similarities with studies conducted in other regions. Highlight the contributions of local research in Pakistan and its potential impact on patient care and management.

Response: The discussion has been revised too add comparative studies with other developing countries like India and Sudan. We have discussed the clinical impact of local studies and identified areas in patient management where there is a paucity of data and further research needs to be done (e.g. experience with novel targeted agents, need for immunoglobulin replacement therapy, use of next generation sequencing etc).

Comment 4: Acknowledge the limitations of the review, such as the small sample sizes of some studies, potential biases, and the absence of certain types of research.

Response: The manuscript has been revised to add a paragraph discussing the limitations.

Comment 5:  Conclude the Discussion by suggesting future research areas that could expand on the existing findings and fill the gaps in knowledge about CLL in Pakistan. This could include investigating specific genetic factors, treatment outcomes, and potential strategies for improving patient care in the country.

Response: The Conclusion has been revised to add the recommended points as relevant.

Thank you again for considering and reviewing our manuscript. Please do not hesitate to reach out to me in case further revisions or clarifications are needed.

Regards,

Mohammad Ammad Ud Din, MD

Round 2

Reviewer 2 Report

no more comments